# Diesel Exhaust Particulates Induce Neutrophilic Lung Inflammation by Modulating Endoplasmic Reticulum Stress-Mediated CXCL1/KC Expression in Alveolar Macrophages

**DOI:** 10.3390/molecules25246046

**Published:** 2020-12-21

**Authors:** Dong Im Kim, Mi-Kyung Song, Hye-In Kim, Kang Min Han, Kyuhong Lee

**Affiliations:** 1National Center for Efficacy Evaluation of Respiratory Disease Products, Korea Institute of Toxicology, 30 Baehak1-gil, Jeongeup 56212, Korea; dongim.kim@kitox.re.kr (D.I.K.); mikyung.song@kitox.re.kr (M.-K.S.); hyein.kim@kitox.re.kr (H.-I.K.); 2Department of Human and Environmental Toxicology, University of Science & Technology, Daejeon 34113, Korea; 3Department of Pathology, Dongguk University Ilsan Hospital, 27, Dongguk-ro, Ilsandong-gu, Goyang-si 10326, Korea; kiekie53@hanmail.net

**Keywords:** particulate matter 2.5, diesel exhaust particulate, endoplasmic reticulum stress, neutrophilic lung inflammation, chemokine CXCL1/KC, alveolar macrophages

## Abstract

Diesel exhaust particulates (DEP) have adverse effects on the respiratory system. Endoplasmic reticulum (ER) abnormalities contribute to lung inflammation. However, the relationship between DEP exposure and ER stress in the respiratory immune system and especially the alveolar macrophages (AM) is poorly understood. Here, we examined ER stress and inflammatory responses using both in vivo and in vitro study. For in vivo study, mice were intratracheally instilled with 25, 50, and 100 μg DEP and in vitro AM were stimulated with DEP at 1, 2, and 3 mg/mL. DEP increased lung weight and the number of inflammatory cells, especially neutrophils, and inflammatory cytokines in bronchoalveolar lavage fluid of mice. DEP also increased the number of DEP-pigmented AM and ER stress markers including bound immunoglobulin protein (BiP) and CCAAT/enhancer binding protein-homologous protein (CHOP) were upregulated in the lungs of DEP-treated mice. In an in vitro study, DEP caused cell damage, increased intracellular reactive oxygen species, and upregulated inflammatory genes and ER stress-related BiP, CHOP, splicing X-box binding protein 1, and activating transcription factor 4 expressions in AM. Furthermore, DEP released the C-X-C Motif Chemokine Ligand 1 (CXCL1/KC) in AM. In conclusion, DEP may contribute to neutrophilic lung inflammation pathogenesis by modulating ER stress-mediated CXCL1/KC expression in AM.

## 1. Introduction

Air pollution in urban areas is associated with adverse effects on the respiratory system [1,2]. Major air pollutants include particulate matter (PM), nitrogen oxides (NOX such as nitrogen dioxide (NO_2_)), ozone (O_3_), sulfur dioxide (SO_2_), carbon monoxide (CO), and hydrocarbons (HC) [3,4]. PM is a complex mixture of suspended solid and/or liquid organic and inorganic substances. It comprises coarse particles (diameter range: 2.5–10 µm (PM10)), fine particles (diameters <2.5 µm (PM2.5)), and ultrafine particles (diameters <0.1 µm (UF)) [2,3]. The physical and biological properties of PM vary with season, region, and source [4,5]. Diesel exhaust particles (DEP) are emitted by diesel vehicles and constitute a major source of PM2.5 [6]. PM2.5 easily penetrate the lung barrier and enter circulation as they are very small. They are considered a major environmental health risk factor [1,7,8]. Several have studies reported that increased exposure to PM2.5 is associated with high respiratory and cardiovascular disease morbidity and mortality. Zanobetti et al. reported that the prevalence of respiratory diseases increased by 2.07% and hospitalization rose by 8% in response to a 10 µg m^−3^ daily increase in PM2.5 [9,10,11,12]. A recent study in China showed that when the PM2.5 concentration rose by 10 µg m^−3^, lung cancer mortality increased by 5.2% [13]. Prior research has already analyzed and identified the association between PM2.5 and respiratory diseases. However, the specific mechanism of the etiological relationship between them has not been fully elucidated.

The endoplasmic reticulum (ER) is an organelle that participates in the biosynthesis, correct folding, and posttranslational modifications of secretory and membrane proteins in cells [14]. However, oxidative and metabolic stress, glucose starvation, elevated protein synthesis, and unfolded and misfolded protein accumulation in the ER lumen induces ER stress, triggering a cellular unfolded protein response (UPR) [15,16]. When ER stress is detected, bound immunoglobulin protein (BiP) dissociates from the ER-transmembrane stress sensors PKR-like eukaryotic initiation factor 2a kinase (PERK), inositol-requiring enzyme 1 (IRE1), and activating transcription factor 6 (ATF6) to restore ER homeostasis. However, when these attempts to recover equilibrium fail, programmed cell death is initiated via CCAAT/enhancer binding to protein-homologous protein (CHOP) upregulation [17]. Several studies have demonstrated that ER stress influences the development and progression of respiratory diseases such as asthma, fibrosis, and acute lung injury [18,19,20,21]. Though ER stress is involved in the pathogenesis of various respiratory disorders, its role in DEP-induced lung inflammation is unclear.

In this study, we investigated the roles and mechanisms of ER stress in DEP-induced lung inflammation by analyzing the cellular changes in bronchoalveolar lavage fluid (BALF), the histological changes, and the ER stress marker (BiP and CHOP) expression levels in the lung tissues of DEP-treated mice. We also evaluated cytotoxicity, oxidative stress, and inflammation- and ER stress-related gene expression in DEP-stimulated AM.

## 2. Results

### 2.1. Changes in Body and Lung Weights

The body weights remained constant throughout the experimental period and did not differ among groups (Figure 1A). After 9 d, the relative mouse lung weights (Figure 1B) gradually increased in a DEP dose-dependent manner and were significantly higher for the DEP 50 and DEP 100 groups than the vehicle control.

### 2.2. DEP Induces Neutrophilic Lung Inflammation in Mice

DEP exposure adversely affects the respiratory system and is a risk factor for respiratory mortality [1,2,3]. To determine whether DEP induces an inflammatory response in mouse lungs, we identified the number of inflammatory cells and analyzed inflammatory cytokine levels in the BALF of DEP-treated mice. The numbers of total cells, macrophages, neutrophils, and lymphocytes in the BALF gradually increased with DEP concentration relative to those in the BALF of the mice in the vehicle control group (Figure 2A). The inflammatory cells in the BALF of the DEP 100 group were significantly higher than those in the BALF vehicle control group mice. Diff-Quik BALF cell staining (Figure 2A) and lung tissue hematoxylin and eosin (H&E) staining (Figure 2B) showed that DEP exposure induced the accumulation of black, particle-laden AM and the infiltration of inflammatory cells in the peribronchiolar, perivascular, and interstitial regions. These results were confirmed by histological scoring of the infiltrated AM and inflammatory cells (Table 1). Additionally, the levels of inflammatory cytokines including TNF-α, IL-1β, and IL-17 were significantly increased in the BALF of the DEP 100 group (Figure 2C–E). Thus, DEP exposure induces neutrophilic lung inflammation in mice and AM might play a crucial role in this disease process.

### 2.3. DEP Initiates ER Stress in DEP-Treated Mice

Endoplasmic reticulum (ER) stress contributes to respiratory disease pathogenesis and plays an important role in inducing lung inflammation [14,15,16,17,18,19,20,21]. To investigate whether DEP exposure is associated with ER stress in our murine model of DEP-induced neutrophilic lung inflammation, we used Western blotting to measure the BiP and CHOP protein expression in lung tissues of DEP-treated mice. ER stress markers were significantly higher in the DEP 100 group lung tissues than in those of the others (Figure 3).

### 2.4. DEP Exposure Induces ER Stress and Activates the UPR Pathway in AM

In an in vivo study, we dominantly observed DEP-laden AM and inflammatory cell infiltrations in lung tissues from DEP-treated mice. Therefore, we studied the association of ER stress and neutrophilic inflammatory responses using a DEP-stimulated AM model and examined the molecular basis for DEP toxicity. In an in vitro study, we confirmed the ER stress markers (including BiP and CHOP) and the UPR pathway (including sXBP-1 and ATF4) in DEP-stimulated AM by RT-qPCR. Relative to the control, the ER stress and UPR pathway-related genes were gradually upregulated in the DEP-stimulated AM. The expression levels of the aforementioned genes were significantly higher in the 3 mg/mL DEP AM than in the untreated control AM (Figure 4). Protein levels of BiP and CHOP were confirmed by Western blotting (Figure 5).

### 2.5. DEP-Stimulated AM Present with Inflammatory Responses

We investigated whether DEP exposure induces inflammatory responses in DEP-stimulated AM. RT-qPCR showed that DEP upregulated the inflammatory genes TNF-α, IL-1β, IL-6, KC, IFN-γ, and TLR4 in AM relative to the control group (Figure 6). The IL-6, KC, IFN-γ, and TLR4 mRNA levels were significantly higher in the DEP-stimulated AM than the control (unchallenged) AM. Protein levels of TNF-α, IL-1β, TLR4, and CXCL1/KC were confirmed by Western blotting (Figure 7).

### 2.6. Effects of Cell Cytotoxicity and Oxidative Stress on DEP Stimulation in AM

DEP or PM exposure induces cell damage via reactive oxygen species (ROS)-mediated oxidative stress [5,22,23]. To determine whether oxidative stress mediates DEP-induced ER stress and inflammatory responses, we evaluated cytotoxicity and ROS production via 3-(4,5 dimethylthiazol-2-thiazyl)-2,5-diphenyl-tetrazolium bromide (MTT) assay and 2’-7’-Dichlorodihydrofluorescein diacetate (DCF-DA) fluorescence intensity measurement, respectively. Compared to the untreated control, DEP lowered cell viability and gradually increased ROS production in the AM. Cytotoxicity was accompanied by significant ROS production in AM challenged with 3 mg/mL DEP (Figure 8A,B). Antioxidant *n*-acetyl-L-cysteine (NAC) pretreatment significantly decreased DEP-increased ROS production in AM (Figure 8C). These results indicate that DEP induces ROS-mediated cytotoxicity in AM.

## 3. Discussion

We can summarize the key findings of this study as follows: (i) DEP induces neutrophilic lung inflammation in in vivo study; (ii) DEP increased ER stress marker expressions of BiP and CHOP in lung tissues of in vivo study; (iii) DEP-laden AM is observed in lung tissues of in vivo study exposed to DEP; (iv) DEP increases ROS-mediated inflammation and ER stress-related expressions in cultured AM; (v) CXCL/1/KC is also released by DEP in cultured AM. (vi) all of these findings suggests that DEP might, at least in part, induce neutrophilic lung inflammation by ER stress-mediated CXCL1/KC expression.

PM is formed mainly by industrial processes and vehicular traffic sources (gasoline and DEP), cooking (coal and oil fuel combustion), and farming and road construction activities [1,2,3,4]. PM2.5 is one of the most harmful air pollutants, and DEP is a major PM2.5. Previously, in a short-term exposure study, we reported that DEP induces lung inflammation by modulating the nuclear factor-κB (NF-κB) signaling pathway in a DEP-treated murine model and DEP-stimulated AM [22]. This result is similar to PM exposure-induced pathogenesis of the lung in humans [11]. However, to date, despite efforts to elucidate the molecular mechanisms associated with the induction of inflammatory respiratory diseases in response to PM2.5 exposure, the molecular mechanisms involved are still not completely understood.

Our results showed that there were no significant changes in body weight between the DEP-instilled group and the vehicle control group. However, the relative mouse lung weights gradually increased in a DEP dose-dependent manner and were observed, especially, in DEP 50 and DEP 100 groups than the vehicle control. In fact, increased lung weight indicates a significant increase in pulmonary vascular permeability and infiltrates inflammatory cells into damaged lung regions [24]. This result may be related to increased types of inflammatory cells, including macrophages, neutrophils, and lymphocytes, in BALF cells of in vivo study. Especially, neutrophils were dominantly infiltrated in the BALF of mice. Our results were also observed in histopathological assessment that infiltrations of inflammatory cells and the number of DEP-laden AM were significantly increased in the lungs of DEP 25, DEP 50, and DEP 100 groups. Moreover, our in vivo study showed that levels of inflammatory cytokines including TNF-α, IL-1β, and IL-17 in BALF were significantly increased. Therefore, we assumed that AM plays an important role in neutrophilic lung inflammation. In the present study, we examined the molecular mechanisms of DEP-induced neutrophilic lung inflammation in a DEP-induced murine lung inflammation model (in vivo study) and in DEP-stimulated AM (in vitro study).

The endoplasmic reticulum (ER) is a specialized organelle that plays crucial roles in protein biosynthesis, protein folding, and posttranslational secretory and membrane protein alterations. ER stress is triggered by certain conditions, such as accumulation of unfolded/misfolded proteins in the ER lumen [15]. Cells initiate an adaptive UPR to restore and maintain ER homeostasis. The four reactions involved in UPR, including stimulation of protein degradation, inhibition of translation, production of chaperones, and cell death, are initiated and activated by the transmembrane protein-mediated signaling pathways IRE1, PERK, and ATF6 [25,26,27,28]. When these protein sensors recognize enhanced ER stress, they activate the UPR process, such as increases in the expression of BiP, a prominent ER-resident chaperone, and CHOP, an apoptotic transcriptional factor induced in response to ER stress. The UPR initially aims to restore proteostasis, but induces cell death under prolonged or severe ER stress [29]. Several studies have demonstrated that ER stress and UPR activation are associated with human pathogenesis and especially inflammation in various metabolic, cardiovascular, and respiratory disorders. Environmental triggers such as cigarette smoking, diesel exhaust, and allergens induce ER stress and UPR in the development and progression of lung diseases including pulmonary fibrosis, asthma, and chronic obstructive pulmonary disease (COPD) [30,31,32,33,34,35,36].

To check whether DEP induces lung injury and to determine whether ER stress contributes to DEP-induced lung injury in vivo, we assessed the cellular changes and inflammatory cytokine levels in LF and histological changes in lung tissues and measured the ER stress marker (BiP and CHOP) expression levels in lung tissue from DEP-treated mice. The number of inflammatory macrophages and neutrophils gradually increased with DEP concentration in the BALF of DEP-treated mice. Moreover, the inflammatory cytokine levels, including TNF-α, IL-1β, and IL-17, were significantly increased in BALF of DEP-treated mice. TNF-α and IL-1β are pro-inflammatory cytokines in the early stage of inflammation [37]. IL-17 is a particular regulator of neutrophil recruitment [38]. Additionally, histological examination showed that the number of neutrophils and the quantity of DEP-pigment AM increased in the lungs of DEP-treated mice relative to those of the untreated control. These findings indicate that DEP induces acute lung inflammation in the early stage and that AM and neutrophils play important roles in this pathological process. The levels of the ER stress markers BiP and CHOP were significantly higher in lung tissues from a murine model of DEP-induced neutrophilic inflammation than the control. Thus, there may have been an interaction between ER stress and neutrophilic lung inflammation pathogenesis in the DEP-treated mice. Therefore, we investigated the association between ER stress and DEP-induced neutrophilic lung inflammation pathogenesis as well as their molecular mechanisms via in vitro DEP stimulation of AM model. We evaluated viability, intracellular ROS production, and inflammation- and ER stress-related expressions in DEP-stimulated AM.

In fact, the primary biological targets of inhaled DEP are cells of the pulmonary epithelium and resident macrophages [39]. Airway epithelium is the first line of defense of the respiratory system against environmental stimuli. Respiratory exposure to PM, including DEP, causes airway epithelial cell damage according to deposits on bronchial epithelium and induces toxicity effects on airways and lung disease through releasing various cytokines including IL-8 and granulocyte-macrophage colony-stimulating factor (GM-CSF), IL-1β, IL-6, IL-11, TNF-α, regulated on activated, normal T cell expressed and secreted (RANTES), intercellular cell adhesion molecule 1 (ICAM-1), vascular cell adhesion molecule 1 (VCAM-1), and E-selectin [40,41]. Our results showed that inflammatory cytokine levels, including TNF-α, IL-1β, and IL-17, were increased by DEP instillation in BALF obtained from in vivo study. Therefore, lung epithelial cells might be damaged by DEP instillation. However, in our in vivo study, deposed DEP were not observed on epithelial cells and the increased infiltration of DEP-laden AM cells was dominantly observed.

Macrophages are major participants in host defense. As constituents in the innate immune system, they engulf and digest pathogens [42]. In the respiratory system, AM keep the air spaces clear by engulfing foreign materials and minimizing the exposure of other airway cells to these substances [42,43]. PM-pigmented AM participates in inflammatory responses by releasing mediators such as IL-6, IL-8, and TNF [23,44] and producing ROS to kill microorganisms [45,46]. Excess ROS resulting from redox imbalances induces oxidative stress, is closely associated with ER stress and UPR activation, and initiates inflammatory processes [31,45,46,47]. Oxidative stress perturbs the redox environment inside the ER where secretory pathway proteins are produced. Consequently, these proteins are misfolded and trigger ER stress. Conversely, other protein misfolding triggers in the ER provoke excessive ROS production. Therefore, oxidative stress and ER stress are intimately involved in inflammation [16]. Our in vitro study confirmed that DEP stimulation significantly upregulated the mRNA or protein levels of ER stress markers BiP and CHOP, the UPR pathway, sXBP-1, and ATF4 in DEP-stimulated AM. Moreover, DEP stimulation increased the mRNA levels of TNF-α, IL-1β, IL-6, IFN-γ, TLR4, and CXCL1/KC in AM. We also confirmed the protein levels of TNF-α, IL-1β, TLR4, and CXCL1/KC in AM. However, the mRNA and protein levels of TNF-α did not significantly increase. One potential limitation of these results is that AM was exposed to DEP in serum-containing media. Okeson et al. has shown that fetal bovine serum can interact with some components of PM [48]. However, because serum is important for proper cell growth and protein production [49], serum-contained media was used, despite the potential for a decrease in the observed toxicity of PM. Additionally, we used MTT assays and DCF-DA fluorescence intensity measurements to assess whether oxidative stress mediates DEP-induced ER stress and inflammatory responses. Our in vitro study revealed that DEP significantly increased cytotoxicity and oxidative stress in AM relative to the control. Moreover, DEP-increased ROS production was significantly decreased by pretreatment of antioxidant NAC in AM. These results indicate that DEP induces ROS-mediated cytotoxicity in AM. Therefore, DEP might promote ER stress and induce inflammation by enhancing oxidative stress in AM.

Several studies demonstrated that after microbial challenges, tissue-resident macrophages are activated to produce neutrophil chemoattractants such as CXCL1, CXCL2, IL-1α, and monocyte chemoattractant protein-1 (MCP-1) [50,51,52]. They induce rapid neutrophil influx to the infection site. These responses are the result of interactions between neutrophils and macrophages. Chemokine CXCL1/KC plays an important role in inflammation as it is a major chemoattractant responsible for recruiting neutrophils. Zhao et al. showed that cellular stress amplifies TLR3/4-induced CXCL1/2 gene transcription in mononuclear phagocytes via RIPK1 [53]. Similar to these results, we showed that the mRNA and protein levels of TLR4 and chemokine CXCL1/KC was significantly upregulated in response to DEP stimulation in AM. Neutrophil-recruiting chemokines, such as Gro (KC, CXCL1), LIX (CXCL5) and MIP2 (CXCL2) are related to IL-17, a regulator of neutrophil recruitment. Studies [14,34] have reported that ER stress contributed to neutrophil inflammation in response to 4-PBA treatment and IL-17A neutralization in the animal models of acute lung injury and severe asthma. Several researchers have proposed various molecular mechanisms such as enhancement of the proinflammatory factors c-Jun N-terminal kinase/Activator protein-1, NF-κB, and phosphoinositide 3-kinases, phosphorylation of double-stranded RNA-activated protein kinase, and oxidative stress-mediated signaling in ER stress and the UPR pathway associated with lung diseases [30,31,32,33,34,35,36]. There are numerous contributing factors connected to the pathogenesis of DEP-induced neutrophilic lung inflammation. We do not have direct evidence to link with chemokine CXCL1/KC released in DEP-stimulated AM and the ER stress-mediated neutrophilic lung inflammation in our model, but our results showed that the mRNA and protein levels of chemokine CXCL1/KC were upregulated in response to DEP stimulation in cell-cultured AM and DEP significantly increased IL-17 protein level in the BALF of an in vivo study. Additionally, our group confirmed CXCL1/KC expression in BALF of DEP-instilled C57BL/6 mice. Despite the different strain, we confirmed that DEP induces neutrophil-dominant inflammatory responses by measuring inflammatory cells and protein levels in BALF and analyzing H&E stain sections in lung tissues of DEP-treated C57BL/6 mice (not shown). Thus, our results suggest that chemokine CXCL1/KC released in DEP-stimulated AM might, at least in part, influence neutrophilic lung inflammation.

## 4. Materials and Methods

### 4.1. Animals

Female Balb/c mice (Orient Bio, Seongnam, Korea) weighing 16.10 ± 0.52 g were housed in a temperature-controlled room (22 ± 3 °C) under a 12 h:12 h light/dark cycle, had ad libitum access to standard laboratory chow and tap water, and were acclimated for 7 d before the experiments. During this time, they presented with normal weight gain and had no adverse clinical signs. All animal experiments were performed in accordance with protocols approved by the Institutional Animal Care and Use Committee of the Korea Institute of Toxicology (No. 1711-0434 and 1801-0009). The Pristima System (v. 7.3; Xybion Medical Systems Corp., Morris Plains, NJ, USA) was used to assign the mice randomly into five weight-matched experimental groups (*n* = 5 per group): naive control, vehicle control, DEP 25, DEP 50, and DEP 100.

### 4.2. DEP Instillation

The mice in the naïve control group received no treatment for the entire experiment. The mice in the vehicle control group received 50 μL saline containing 0.05% (*v*/*v*) Tween 80 (Sigma-Aldrich Corp., St. Louis, MO, USA). The mice in the DEP 25, DEP 50, and DEP 100 groups were intratracheally instilled with 25 μg, 50 μg, and 100 μg DEP (SRM 2975; National Institute of Standards and Technology, Gaithersburg, MD, USA) dispersed in a 50 μL vehicle on days 1, 4, and 7, respectively [22]. During the in vivo experiment, the total amount of exposure is 75, 150, and 300 μg per mouse. The mice in the DEP 25, DEP 50, and DEP 100 groups were exposed with each delivered dose of 1.4, 2.9, and 5.9 mg/kg DEP daily in mice. However, if the delivered dose was calculated during the experimental period, then the delivered doses in the in vivo study were 4.3–17.2 mg/kg DEP. Based on the exposure duration per day (i.e., 6 h/day), the respiratory minute volume (19.9 L/min), human body weight (60 kg), and proportion by weight of particles that are inhalable by humans (IF = 0.5), the delivered dose of DEP for humans is 0.0034 mg/kg [54]. Although the delivery dose of DEP is somewhat high, considering uncertainty factors such as limitations of the study, animal-to-human extrapolation, sensitive sub-populations, and inadequacies of databases, the doses of DEP used in the present in vivo study are within a sufficiently reasonable range.

### 4.3. Body and Lung Weight Measurements

The body weight of mice was measured on days 0, 1, 4, 7, and 9. On day 9, the mice were sacrificed and their lungs were weighed.

### 4.4. BALF Preparation

At 48 h after the last DEP instillation, the mice were anesthetized with isoflurane and exsanguinated. Their left lungs were ligated and their right lungs were gently lavaged thrice via a tracheal tube using a total volume of 0.7 mL phosphate-buffered saline (PBS). The cells in the BALF were counted with a NucleoCounter (NC-250; ChemoMetec, Gydevang, Denmark). To differentiate the cell types, BALF cell smears were prepared with Cytospin (Thermo Fisher Scientific, Waltham, MA, USA) and stained with Diff-Quik solution (Dade Diagnostics, Aguada, Puerto Rico). Two hundred cells per slide were counted.

### 4.5. Measurement of Cytokine and Chemokine Levels in BALF

TNF-α, IL-1β, IL-6, IL-17, IL-33, IFN-γ, MCP-1, and KC levels in BALF were quantified using Meso Scale Discovery (MSD) Sector Imager 2400A (Rockville, MD, USA).

### 4.6. Histological Analysis

On day 9 after the first DEP administration, the mice were sacrificed for histological analysis. Lung tissues were excised, fixed in 10% (*v*/*v*) neutral-buffered formalin, dehydrated, embedded in paraffin, sliced into 4-μm sections, deparaffinized with xylene, stained with hematoxylin and eosin (H&E; Sigma-Aldrich Corp., St. Louis, MO, USA), and viewed under a light microscope (Axio Imager M1; Carl Zeiss AG, Oberkochen, Germany). The degree of inflammation was rated on a 0–4 scale as previously described [55,56].

### 4.7. Protein Extract Preparation and Western Blot Analysis

Lung tissues were homogenized and lysed in radioimmunoprecipitation (RIPA) buffer (Thermo Fisher Scientific, Waltham, MA, USA) with a protease inhibitor cocktail. The protein concentrations were determined with Bradford reagent (Bio-Rad Laboratories, Hercules, CA, USA). The samples were loaded onto SDS-PAGE gel. After electrophoresis at 120  V for 90 min, the proteins were wet-transferred to polyvinylidene difluoride (PVDF) membranes (Bio-Rad Laboratories, Hercules, CA, USA) at 250  mA for 90  min. Nonspecific sites were blocked with 5% (*w*/*v*) nonfat dry milk in Tris-buffered saline plus Tween 20 (25  mM Tris (pH 7.5), 150  mM NaCl, and 0.1% (*v*/*v*) Tween 20) for 1 h and the blots were incubated overnight at 4 °C with anti-BiP (Cell Signaling Technology, Beverly, MA, USA), anti-CHOP (Cell Signaling Technology, Beverly, MA, USA), TNF-α (Santa Cruz Biotechnology, Dallas, TX, USA), IL-1β, TLR4 system (Thermo Fisher Scientific, Waltham, MA, USA), KC (Biovision Inc., Milpitas, CA, USA), and anti-actin (Santa Cruz Biotechnology, Dallas, TX, USA) antibodies. The latter was a reference protein. Anti-rabbit or anti-mouse horseradish peroxidase-conjugated immunoglobulin G (Cell Signaling Technology, Beverly, MA, USA) was used to detect antibody binding. Specific antibody binding was visualized with an iBright CL1000 imaging system (Thermo Fisher Scientific, Waltham, MA, USA) after treatment with enhanced chemiluminescence (ECL) system reagents (Thermo Fisher Scientific, Waltham, MA, USA). Densitometry was performed on each relative band intensity with iBright CL1000 image software (Thermo Fisher Scientific, Waltham, MA, USA). To quantify specific bands, squares of the same size were drawn around all bands, the densities were measured, and the values were adjusted according to the background densities near the bands. The densitometry data were expressed as the relative ratios of the target proteins to the reference protein. The relative ratios of the target control group proteins were arbitrarily set to unity.

### 4.8. Murine AM Culture and DEP, H_2_O_2_, and NAC Treatment

The MH-S murine AM cell line (CRL-2019) was purchased from the American Type Culture Collection (Manassas, VA, USA) and maintained in Roswell Park Memorial Institute (RPMI) 1640 medium (Gibco, Grand Island, NY, USA) supplemented with 10% (*v*/*v*) fetal bovine serum (FBS) and 1% (*w*/*v*) penicillin-streptomycin solution (Gibco, Grand Island, NY, USA). Cells were seeded at a concentration of 2 × 10^5^ in 500 μL of medium using sterile 24-well culture plates. The cells were incubated at 37 °C under a humidified 5% CO_2_ atmosphere and treated with DEP (1 mg/mL, 2 mg/mL, and 3 mg/mL). In in vitro study, AM was treated with DEP (1, 2, and 3 mg/mL). After 3 h, the cells were analyzed to investigate the response of AM by DEP in the early stages of inflammation. Furthermore, AM was treated with H_2_O_2_ (Sigma-Aldrich Corp., St. Louis, MO, USA; 10 mM for 30 min) as a positive control for ROS and was pretreated with antioxidant NAC (Sigma-Aldrich Corp., St. Louis, MO, USA; 5 mM for 2 h).

### 4.9. Cell Viability and Reactive Oxygen Species (ROS) Measurements

DEP cell viability and ROS production were evaluated with MTT (Sigma-Aldrich Corp., St. Louis, MO, USA) and 3.3 μM DCF-DA (Thermo Fisher Scientific, Waltham, MA, USA), respectively. For the cell viability determination, after DEP stimulation, a 1 mg/mL MTT solution was added to the cells and they were then incubated at 37 °C for 3 h. The supernatant was removed and the formazan crystals were dissolved in 100 μL dimethyl sulfoxide (DMSO; Junsei Chemical Co., Tokyo, Japan). Absorbances were measured at 570 nm and the background control was measured at 690 nm in a microplate reader (BioTek, Winooski, VT, USA). Cell viabilities were reported as percentages of the optical density values. To determine the ROS levels, the AM were incubated for 30 min at 37 °C in complete RPMI 1640 medium containing 3.3 μM DCF-DA. The DCF-DA intensities in the cells were immediately measured at 495 nm (excitation) and 529 nm (emission) in a microplate reader. The ROS production levels in the cells were represented as percentages of the DCF-DA intensities relative to the cell viabilities in each well. Moreover, the DCF-DA intensities in the live cells were performed using Cytoflex flow cytometry (Beckman coulter, Brea, CA, USA). The control group was defined as 100%.

### 4.10. RNA Extraction and Quantitative Real-Time Polymerase Chain Reaction (RT-qPCR)

RNA was extracted from the AM with the RNeasy Mini kit (Qiagen, Venlo, The Netherlands) according to the manufacturer’s protocol and was quantified by measuring absorbance at 260 nm. Then, 1 μg RNA was reverse-transcribed to cDNA in an ImProm-II reverse transcription system (Promega, Madison, WI, USA). The cDNA was quantified with a QuantStudio5 Real-Time PCR system and SYBR Green PCR Master Mix (Applied Biosystems, Foster City, CA, USA). The 20-μL reaction mixtures consisted of 10 μL SYBR Green PCR Master Mix, 0.5 pmol of each primer, and 0.5 μL undiluted first-strand cDNA. The cycling conditions were as follows: 95 °C for 10 min followed by 40 cycles of 95 °C for 15 s and 59 °C for 1 min. The cycle threshold method was used to calculate the relative changes in target gene expression in QuantStudio Design and Analysis v. 1.4 software (Applied Biosystems, Foster City, CA, USA). The target gene expression levels were normalized to that of actin and were expressed as fold changes. The normalized values of the target gene expression levels in the control group were set to unity. The PCR primers are listed in Table 2.

### 4.11. Statistical Analysis

The data were statistically processed with SigmaPlot v. 12 software (Systat, San Jose, CA, USA). Data are means ± SD. Statistical comparisons were performed by one-way ANOVA followed by Dunnett’s test. *p* < 0.05 was considered statistically significant.

## 5. Conclusions

Our in vivo study showed that ER stress markers increased in DEP-induced neutrophilic lung inflammation. DEP induced ROS production and ER stress- and inflammation-related gene expression in AM of in vitro study. Further research is needed to clarify the mechanisms of these processes. The findings herein suggest that DEP exposure triggers ER stress, activates UPR, increases oxidative stress, damages AM, and induces chemokine CXCL1/KC-mediated neutrophilic lung inflammation.

## Figures and Tables

**Figure 1 molecules-25-06046-f001:**
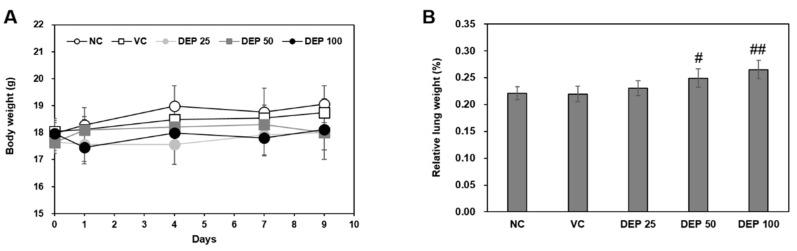
Changes in (**A**) body weight and (**B**) relative lung weight in mice in response to DEP instillation. Relative lung weights were calculated as follows: relative organ weight = lung weight (g)/final body weight (g) × 100%. Data are means ± SD (*n* = 5 per group). ^#^
*p* < 0.05 or ^##^
*p* < 0.01 vs. vehicle control.

**Figure 2 molecules-25-06046-f002:**
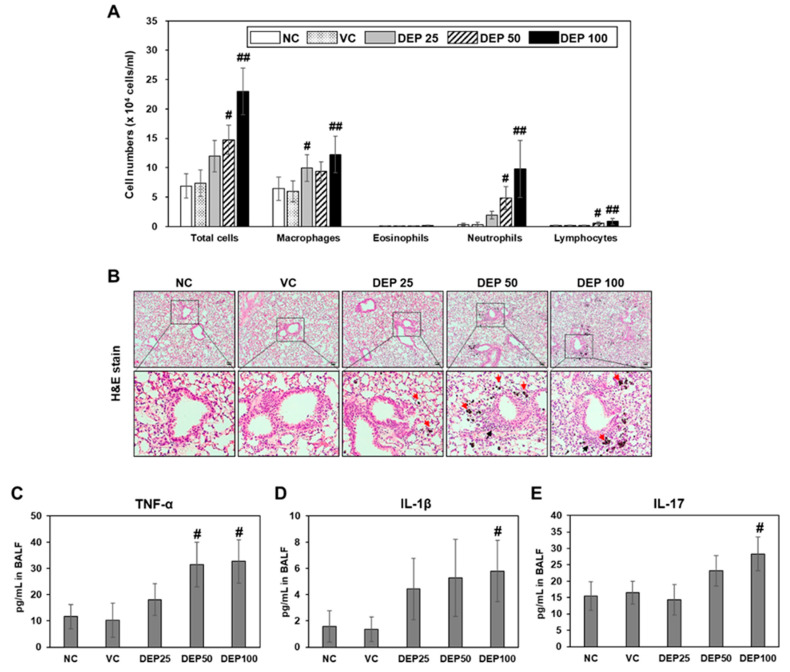
(**A**) Cellular changes in BALF obtained from naïve control (NC), vehicle control (VC), and DEP 25 μg (DEP 25), DEP 50 μg (DEP 50), or DEP 100 μg (DEP 100) mice. Mice were treated 3 times by DEP and sacrificed at day 9. Data are means ± SD (*n* = 5 per group). ^#^
*p* < 0.05 or ^##^
*p* < 0.01 vs. vehicle control. (**B**) Histological changes in lung tissue caused by DEP instillation. Representative H&E-stained sections of lung tissue excised from DEP-induced mice. Red and black arrows indicate black particle-laden alveolar macrophages and inflammatory cells, respectively. Scale bars: 50 μm. Inflammatory cytokine levels including (**C**) TNF-α, (**D**) IL-1β, and (**E**) IL-17 in BALF of mice. Data are means ± SD (*n* = 4 per group). ^#^
*p* < 0.05 vs. vehicle control.

**Figure 3 molecules-25-06046-f003:**
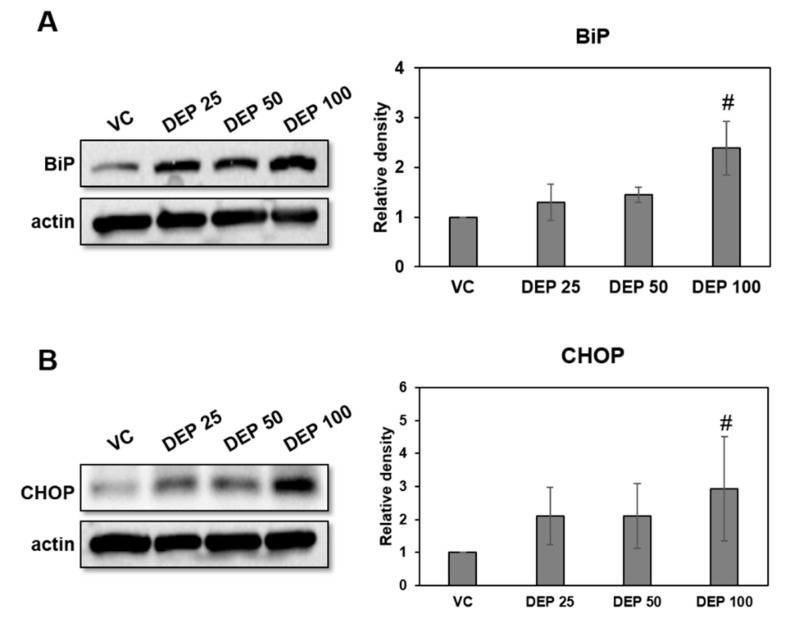
Expression levels of the ER stress markers BiP and CHOP in DEP-induced mice. Representative Western blots and relative densities of (**A**) BiP and (**B**) CHOP in lung tissues of DEP-induced mice. Mice were treated three times by DEP and sacrificed at day 9. Data are means ± SD (*n* = 5 per group). ^#^
*p* < 0.05 vs vehicle control.

**Figure 4 molecules-25-06046-f004:**
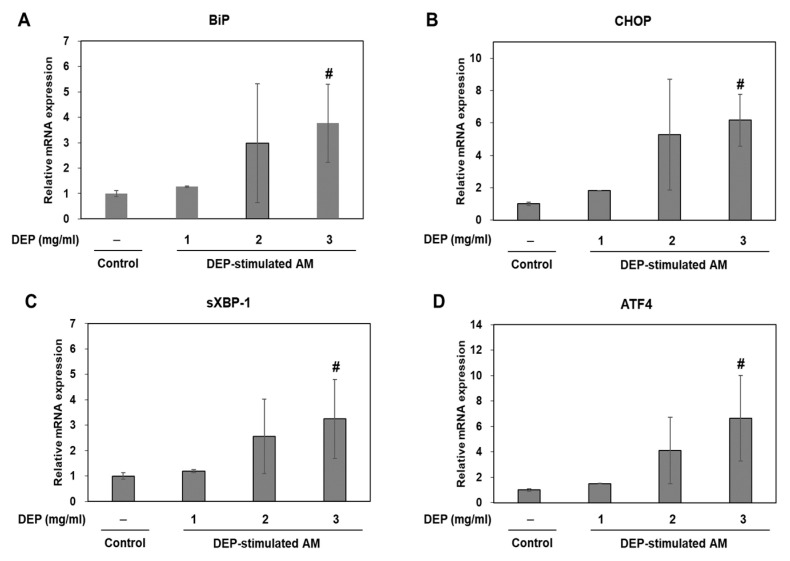
ER stress-related gene expression levels. (**A**) BiP, (**B**) CHOP, (**C**) sXBP-1, and (**D**) ATF4 in DEP-stimulated AM. AM were stimulated DEP 1, 2, or 3 mg mL^−1^ with. RT-qPCR was performed 3 h after DEP stimulation. Data are means ± SD (*n* = 3 per group). ^#^*p* < 0.05 vs. control.

**Figure 5 molecules-25-06046-f005:**
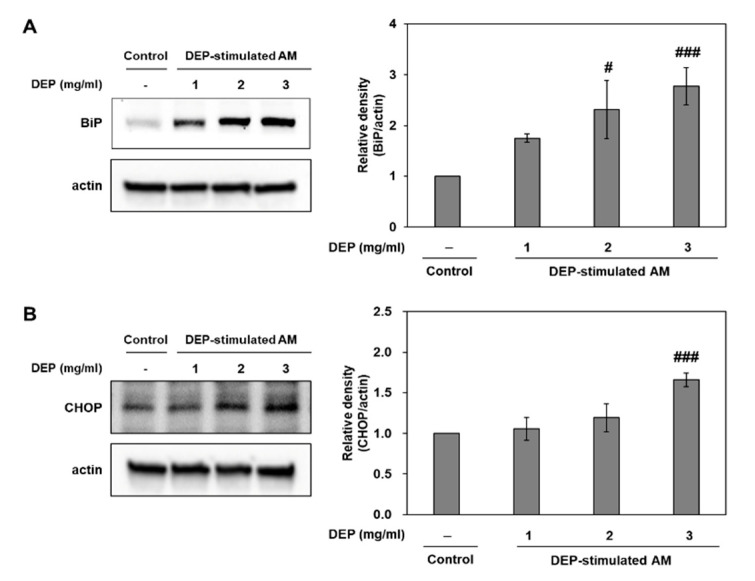
Expression levels of the ER stress markers BiP and CHOP in DEP-stimulated AM. Representative Western blots and relative densities of (**A**) BiP and (**B**) CHOP in DEP-stimulated AM. AM were stimulated with DEP 1, 2, or 3 mg mL^−1^. Western blotting was performed 3 h after DEP stimulation. Data are means ± SD (*n* = 3 per group). ^#^
*p* < 0.05 or ^###^
*p* < 0.001 vs. control.

**Figure 6 molecules-25-06046-f006:**
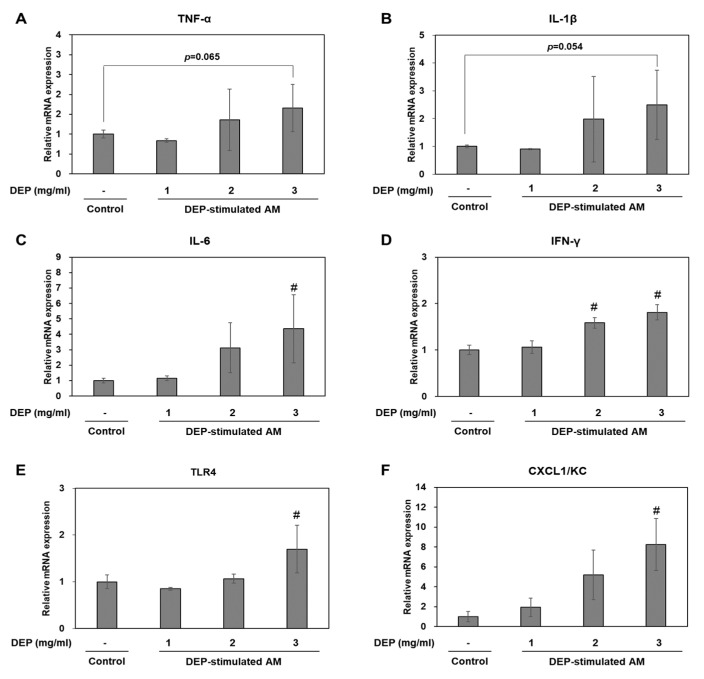
(**A**–**F**) Relative mRNA expression levels for inflammatory factors in DEP-stimulated AM. Data are means ± SD (*n* = 3 per group). ^#^
*p* < 0.05 vs. control.

**Figure 7 molecules-25-06046-f007:**
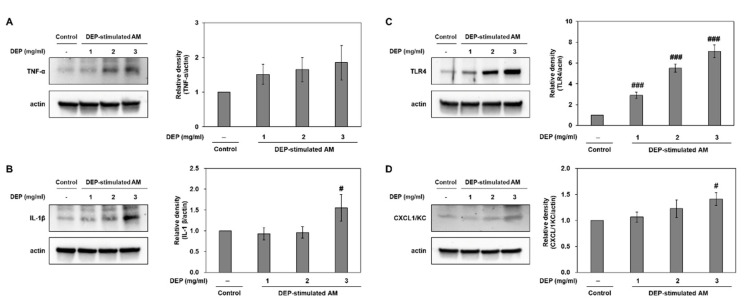
Representative Western blots and relative densities of (**A**) TNF-α, (**B**) IL-1β, (**C**) TLR4, and (**D**) CXCL1/KC in DEP-stimulated AM. AM were stimulated with DEP 1, 2, or 3 mg mL^−1^. Western blotting was performed 3 h after DEP stimulation. Data are means ± SD (*n* = 3 per group). ^#^
*p* < 0.05 or ^###^
*p* < 0.001 vs. control.

**Figure 8 molecules-25-06046-f008:**
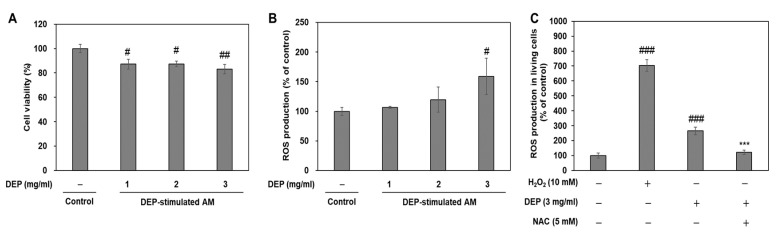
DEP induces oxidative stress-mediated cytotoxicity in DEP-treated AM. (**A**) Cytotoxicity assessment by MTT assay. (**B**) ROS production was measured by DCF-DA staining. AM were subjected to DEP (1, 2, or 3 mg mL^−1^). DCF-DA staining and the MTT assay were performed 3 h after DEP challenge. (**C**) NAC pretreatment inhibits oxidative stress in DEP-stimulated AM. ROS production was measured in living cells by DCF-DA staining using flow cytometry. AM were pretreated with H_2_O_2_ for 30 min and NAC for 2 h and were stimulated with DEP (3 mg mL^−1^) for 3 h. Data are means ± SD (*n* = 3 per group). ^#^*p* < 0.05, ^##^*p*< 0.01, ^###^*p*< 0.001 or vs. control; ^***^*p*< 0.001 or vs. DEP group.

**Table 1 molecules-25-06046-t001:** Histologic scores for lungs of DEP-induced mice.

Group		Naive Control	Vehicle Control	DEP 25	DEP 50	DEP 100
Accumulation of black particle-laden	Minimal	0	0	1	0	0
alveolar macrophages and black	Mild	0	0	4	2	0
particles in alveolar lumen	Moderate	0	0	0	3	2
	Marked	0	0	0	0	3
	Mean ± SD	0	0	1.8 ± 0.45 ^##^	2.6 ± 0.55 ^##^	3.6 ± 0.55 ^##^
Inflammatory cell infiltration,	Minimal	0	0	4	3	0
peribronchiolar/perivascular/interstitial	Mild	0	0	0	2	5
	Mean ± SD	0	0	0.8 ± 0.45 ^#^	1.4 ± 0.55 ^##^	2 ± 0.0 ^##^

Data are means ± SD for five mice per group. DEP, diesel exhaust particulate. ^#^
*p* < 0.05 or ^##^
*p* < 0.001 vs. vehicle control.

**Table 2 molecules-25-06046-t002:** RT-qPCR primers.

Primer Name	Forward Primer	Reverse Primer
BiP	TTCAGCCAATTATCAGCAAACTCT	TTTTCTGATGTATCCTCTTCACCAGT
CHOP	CCACCACACCTGAAAGCAGAA	AGGTGAAAGGCAGGGACTCA
sXBP-1	CTGAGTCCGAATCAGGTGCAG	GTCCATGGGAAGATGTTCTGG
ATF4	GGGTTCTGTCTTCCACTCCA	AAGCAGCAGAGTCAGGCTTTC
TNF-α	ATGAGCACAGAAAGCATGA	AGTAGACAGAAGAGCGTGGT
IL-1β	CAACCAACAAGTGATATTCTCCATG	ATCCACACTCTCCAGCTGCA
IL-6	GCTACCAAACTGGATATAATCAGGA	CCAGGTAGCTATGGTACTCCAGAA
IFN-γ	TTCTTCAGCAACAGCAAGGC	TCAGCAGCGACTCCTTTTCC
TLR4	AAACGGCAACTTGGACCTGA	AGCTTAGCAGCCATGTGTTCCA
CXCL1/KC	CGCTCGCTTCTCTGTGCA	ATTTTCTGAACCAAGGGAGCT
actin	GGCACCACACCTTCTACAATG	GGGGTGTTGAAGGTCTCAAAC

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
