# Peer review of "Diesel Exhaust Particulates Induce Neutrophilic Lung Inflammation by Modulating Endoplasmic Reticulum Stress-Mediated CXCL1/KC Expression in Alveolar Macrophages"

_molecules, 2020, doi:10.3390/molecules25246046_

Round 1
Reviewer 1 Report
The authors try to evaluate the relationship of ER-stress in alveolar Macrophages and diesel exhaust particulates (DEP) induced neutrophilic lung inflammation. In general, this study is interesting and important. However, the experiment designs are inadequate. The authors need to provide more advantageous results and explanations to improve their hypothesis and also for increasing the clarity of this study.
Concerns are listed below:
- In the method section, the DEP instillation part. I want to make sure the received dosage of each mouse. In myunderstanding,a mouse was received three times of DEP (25, 50, and 100 μg), the total exposed doses were 75, 150, and 300 μg, right?
- Please define BALF cells, does it means bronchoalveolar lavage fluid cells?
- How to identify lung inflammation? Any biochemistry biomarkers supported? COX-2, PGE2 signals? Only histological results seem slightly weak.
- The doses were largely different between the in vitro and the in vivo study. In the animal study, an animal has receivedμg range DEP and significant ER-stress responses could be obtained, while AM cell needed to be treated with mg range doses? Is the AM cell really plays important roles in DEP-induced cell toxicities?
- Please provide how much AM cells were used for each experiment.
- CHOP signal has been indicated as a pro-apoptotic event, while GRP78 (Bip) is pro-survival events in several articles. The Unbalance of CHOP and GRP78 causes cell apoptosis. In the current study, the MTT viability assay could not explain that the cell's death was via apoptosis or necrosis after DEP exposure.
- Furthermore, In Figure 6, both cell viability and ROS were determined at 3h. Theoretically, increases in ROS production should precede eventual apoptosis. I wouldn’t expect an increase in ROS production if apoptosis is occurring.
- The results of current study could not be fully linkedbetween the in vivo and in vitro studies. The CHOP and Bip increases in AM cell could not reflect the CHOP and Bip levels of whole lung tissue extract, the results could not indicate the ER-stress in the lung is contributed by AM ER-stress.
- The mRNA increases could not fully reflect the protein expression. Why the CHOP and BIP signals in lung extract were detected by immunoblotting, but just measured the mRNA level in AM cells. Moreover, the AM cells could secrete several cytokines and chemokines, I think the protein levels of these significant signals needed to be measured by immunoblotting and ELISA methods.
- Discussion is very long and unfocused, It seems to me that the manuscript needs a unifying message that is unfortunately diluted by these broad and unfocused observations of previous studies.The sentences of doses selected principles could be partially moved to the method section. More current findings could be discussed in more detail, why the body weight increases? Why lung weight increase? If body weight increased is associated with body inflammation? how about the level of c-reactive protein? TNF-a and IL-1 beta are important factors for neutrophilic lung inflammation, why AM did not raise significant TNF-a and IL-1 beta mRNA expressions? And the roles of CHOP and Bip in the acute and chronic ER-stress.
- Some conclusion is over-explanation, for example, authors mention “Thus, our results suggest that chemokine CXCL1/KC released in DEP-stimulated AM could influence the induction and maintenance of ER stress-mediated neutrophilic lung inflammation.” The current data could not support this hypothesis, due to which lack the level of CXCL1 release from AM after DEP exposure, the author could not demonstrate that neutrophils recruitment is mediated by AM. Moreover, there is no direct evidence to suggest the CXCL1/KC expression is mediated by ER-stress in AM. Hence, I do not fully agreewith the sentence of the conclusion “Nevertheless, the findings of the present study suggest……upregulating ER stress and UPR-mediated CXCL1/KC expression in AM.”.
Author Response
Manuscript ID: molecules-992748
Responses to the Reviewers' Comments
We appreciate very much the editor and the reviewers for the constructive comments. We also thank the editor and the reviewers for the effort and time put into the review of the manuscript. Each comment has been carefully considered point by point and responded. Responses to the reviewers and changes in the revised manuscript are as follows. Thank you for your consideration. I am looking forward to your positive response.
# Reviewer 1: Thank you for your thoughtful and thorough review of our manuscript.
Comments and Suggestions for Authors:
The authors try to evaluate the relationship of ER-stress in alveolar Macrophages and diesel exhaust particulates (DEP) induced neutrophilic lung inflammation. In general, this study is interesting and important. However, the experiment designs are inadequate. The authors need to provide more advantageous results and explanations to improve their hypothesis and also for increasing the clarity of this study.
Concerns are listed below:
- In the method section, the DEP instillation part. I want to make sure the received dosage of each mouse. In my understanding, a mouse was received three times of DEP (25, 50, and 100 μg), the total exposed doses were 75, 150, and 300 μg, right?
Response: It is right. We intratracheally exposed 25, 50, and 100 μg of DEP to mouse at a time and exposed them total 3 times during in vivo experiment. Therefore, the total amount of exposure is 75, 150, and 300 μg per mouse. We added information in 4.2 DEP instillation (page 10, line 325).
- Please define BALF cells, does it means bronchoalveolar lavage fluid cells?
Response: It is right. We first mentioned full name of commented abbreviation in the text (page 2, line 67).
- How to identify lung inflammation? Any biochemistry biomarkers supported? COX-2, PGE2 signals? Only histological results seem slightly weak.
Response: We performed lung histology and differential cell counts to show lung inflammation in in vivo study. As in the references below, many researchers are also checking in the same way as we do. Additionally, we confirmed that DEP exposure increases inflammatory cytokine levels such as TNF-α, IL-1β, and IL-17 in BALF of same in vivo study design. So, we added the result of inflammatory cytokine levels in BALF of in vivo study (page 3, Figure 2C-2E).
Reference
- Joseph Soltzberg, Sarah Frischmann, Christiaan van Heeckeren, Nicole Brown, Arnold Caplan, and Tracey L. Bonfield. Quantitative Microscopy in Murine Models of Lung Inflammation. Anal Quant Cytol Histol. 2011 October ; 33(5): 245–252.
- Zbigniew Zaslona, Sally Przybranowski, Carol Wilke, Nico van Rooijen, Seagal Teitz-Tennenbaum, John J. Osterholzer, John E. Wilkinson, Bethany B. Moore and Marc Peters-Golden, Resident Alveolar Macrophages Suppress, whereas Recruited Monocytes Promote, Allergic Lung Inflammation in Murine Models of Asthma. J Immunol 2014; 193:4245-4253
- The doses were largely different between the in vitro and the in vivo study. In the animal study, an animal has received μg range DEP and significant ER-stress responses could be obtained, while AM cell needed to be treated with mg range doses? Is the AM cell really plays important roles in DEP-induced cell toxicities?
Response: In our in vivo study, DEP-laden AM was significantly observed and expression of ER stress markers (BiP and CHOP) was statistically increased in lung from murine model of DEP-induced neutrophilic lung inflammation. So, we assumed that AM may play an important role in the DEP-induced neutrophilic lung inflammation and ER stress responses. We tried to set in vitro system that can explain the pathological changes observed in in vivo experiment. In fact, when we previously used concentration of 100 μg/ml DEP suspended with saline containing 0.05% Tween 80 in cultured AM, we confirmed that DEP increases ROS and inflammatory cytokine levels and induces cell damage. However, because detergent Tween 80 can influence in cytotoxicity, we tried to find optimal concentrations that DEP induces cytotoxicity without any detergent. Finally, we confirmed that cytotoxicity at the concentration (1, 2, and 3 mg/ml) used in current study was similar to that at the concentration of 100 μg/ml dissolved in the saline containing 0.05% Tween 80 and selected mg range doses in in vitro study.
- Please provide how much AM cells were used for each experiment.
Response: In in vitro study, AM cells were seeded at a concentration of 2×105 in 500 μL of medium using sterile 24-well culture plates. We added information in Materials and Methods section (page 11, line 380-381).
- CHOP signal has been indicated as a pro-apoptotic event, while GRP78 (Bip) is pro-survival events in several articles. The Unbalance of CHOP and GRP78 causes cell apoptosis. In the current study, the MTT viability assay could not explain that the cell's death was via apoptosis or necrosis after DEP exposure.
Response: We agree reviewer’s comment. So, we performed cell death assay by annexin V and PI staining. However, we could not observe apoptosis or necrosis by DEP stimulation in in vitro study. Also, LDH level also did not change in supernatants of DEP-stimulated AM. In current study, we could not directly observe CHOP-mediated apoptosis by DEP stimulation for 3 h. Thus, MTT assay result seems to show that DEP induces cytotoxicity via inhibiting cell proliferation in DEP-stimulated AM.
- Furthermore, In Figure 6, both cell viability and ROS were determined at 3h. Theoretically, increases in ROS production should precede eventual apoptosis. I wouldn’t expect an increase in ROS production if apoptosis is occurring.
Response: Thank you for your comment. I agree reviewer’s comments. So, in our result, DCF-DA intensities were normalized with cell viabilities in each well. Additionally, we checked DCF-DA intensities in live cells using flow cytometry after H2O2 (positive control for ROS) or antioxidant N-acetyl-L-cysteine (NAC) treatment in 3 mg/ml DEP-stimulated AM. Our results showed that NAC pretreatment significantly decreased DEP-increased ROS production in AM. These results indicate that ROS is an important mediator of DEP-induced cytotoxicity in AM. This result was added, and Figure 6 was changed to Figure 8 through revising the paper. (page 7, Figure 8).
- The results of current study could not be fully linked between the in vivo and in vitro studies. The CHOP and Bip increases in AM cell could not reflect the CHOP and Bip levels of whole lung tissue extract, the results could not indicate the ER-stress in the lung is contributed by AM ER-stress.
Response: ER stress response can induce in most cells including structure cells and immune cells and play important role in lung inflammation. So, we examined ER stress and inflammatory responses in DEP-treated in vivo study. As mentioned previously, our results showed that levels of ER stress marker in lung tissues and inflammatory cytokine in BALF were significantly increased by DEP. Additionally, DEP-laden AM was significantly observed in H&E section of lung tissues obtained from in vivo study, and we assumed that AM may play an important role in the DEP-induced inflammation and ER stress responses. Therefore, we explained the phenomenon observed in in vivo experiment using in vitro study of DEP-stimulated AM.
- The mRNA increases could not fully reflect the protein expression. Why the CHOP and BIP signals in lung extract were detected by immunoblotting, but just measured the mRNA level in AM cells. Moreover, the AM cells could secrete several cytokines and chemokines, I think the protein levels of these significant signals needed to be measured by immunoblotting and ELISA methods.
Response: Thank you for your comment. We performed immunoblot analysis of CHOP, BiP, several cytokines (TNF-α, IL-1β and, TLR4), and KC in DEP-stimulated AM (page 5, Figure 5; page 6, Figure 7).
- Discussion is very long and unfocused, It seems to me that the manuscript needs a unifying message that is unfortunately diluted by these broad and unfocused observations of previous studies. i) The sentences of doses selected principles could be partially moved to the method section. ii) More current findings could be discussed in more detail, why the body weight increases? iii) Why lung weight increase? iv) If body weight increased is associated with body inflammation? how about the level of c-reactive protein? v) TNF-α and IL-1 β are important factors for neutrophilic lung inflammation, why AM did not raise significant TNF-α and IL-1 β mRNA expressions? vi) And the roles of CHOP and Bip in the acute and chronic ER-stress.
Response: Thank you for your comment. We revised discussion section in more detail as review’s comments.
- i) We moved the sentences of doses selected principles to the method section as your comment (page 10, line 325-334).
- ii) In our in vivo study, there were no changes on body weight during experiment periods.
iii) But, the relative mouse lung weights gradually increased in a DEP dose-dependent manner and were observed, especially, in DEP 50 and DEP 100 groups than the vehicle control. In fact, increased lung weight indicates a significant increase of pulmonary vascular permeability and infiltrates inflammatory cells into damaged lung regions (1). This result may be related to increase types of inflammatory cells, including macrophages, neutrophils, and lymphocytes, in BALF cells of in vivo study (page 7, line 189-page 8, line 202).
- iv) Relative lung weight increased is associated with lung inflammation. Our in vivo study showed that levels of inflammatory cytokines including TNF-α, IL-1β, and IL-17 in BALF as well as infiltration of inflammatory cells in BALF and lung tissues were significantly increased (page 7, line 189-page 8, line 202).
- v) Also, the mRNA and protein levels of TNF-α did not significantly increase in in vitro One potential limitation of these results is that AM was exposed to DEP in serum-containing media. Okeson et al. has shown that fetal bovine serum can interact with some components of PM (2). However, because serum is important for proper cell growth and protein production (3), serum-contained media was used, despite the potential for a decrease in the observed toxicity of PM (page 9, line 267-271).
- vi) we mentioned about the roles of CHOP and Bip in the acute and chronic ER-stress (page 8, line 210-214).
Reference
- James C. Parker and Mary I. Townsley. Evaluation of lung injury in rats and mice. Am J Physiol Lung Cell Mol Physiol 286: L231–L246, 2004;10.1152/ajplung.00049.2003.
- Okeson, C. D., Riley, M. R., and Riley-Saxton, E. (2004). In Vitro Alveolar Cytotoxicity of Soluble Components of Airborne Particulate Matter: Effects of Serum on Toxicity of Transition Metals. Toxicol. In Vitro,18:673–680.
- Hsiao, I. L., and Huang, Y. J. (2013). Effects of Serum on Cytotoxicity of Nano- and Micro-Sized ZnO Particles. J. Nanopart. Res., 15:1829.
- Some conclusion is over-explanation, for example, authors mention “Thus, our results suggest that chemokine CXCL1/KC released in DEP-stimulated AM could influence the induction and maintenance of ER stress-mediated neutrophilic lung inflammation.” The current data could not support this hypothesis, due to which lack the level of CXCL1 release from AM after DEP exposure, the author could not demonstrate that neutrophils recruitment is mediated by AM. Moreover, there is no direct evidence to suggest the CXCL1/KC expression is mediated by ER-stress in AM. Hence, I do not fully agree with of the conclusion “Nevertheless, the findings of the present study suggest……upregulating ER stress and UPR-mediated CXCL1/KC expression in AM.”.
Response: Thank you for your comment. We added in vivo data of inflammatory cytokine levels (TNF-α, IL-1β, and IL-17) in BALF (page 3, Figure 2C-2E) and in vitro data of ER markers (BiP and CHOP), inflammatory cytokines (TNF-α, IL-1β, and TLR4), and CXCL1/KC proteins (page 5, Figure 5; page 6, Figure 7). We entirely revised discussion section as review’s comments. Also, we deleted a “Nevertheless, the findings of the present study suggest……upregulating ER stress and UPR-mediated CXCL1/KC expression in AM.” in the last sentence of discussion.
Reviewer 2 Report
In this manuscript Kim et al. analyze the ER stress and inflammatory responses in a mouse model of diesel exhaust particulate (DEP) exposure (3 times within 9 days). Moreover, they study the induction of inflammation, ER stress and ROS in a murine alveolar macrophage cell line (MH-S). They show that DEP induces neutrophil recruitment and upregulation of ER stress markers in the lungs of DEP-treated mice. Stimulation of MHS cells with DEP induces the expression of ER stress markers, proinflammatory cytokines (including CXCL-1/KC) and production of ROS. The study is interesting because is focused in the understanding of the mechanisms explaining the adverse effects of air pollution on the respiratory system. However, some results are overestimated and final conclusions need to be supported with additional experimental evidences.
Major points
-Title tends to overestimate data present in the paper. Data do not demonstrate that AM present in the lungs of DEP-treated mice are responsible of the neutrophilic inflammation observed. They do not show neither a link between levels of KC in BALF (not showed) and lung inflammation. Data do not show that KC expression is mediated by ER stress in MH-S cells.
-In line with the previous point, authors should measure levels of KC in the BALF of DEP treated mice in agreement with in vitro data and in order to reinforce the main hypothesis.
-To improve the hypothesis of the study, could be possible to isolate AM from BALF of treated mice and analyze KC and ER stress markers expression?
- Authors focus on the role of AM in DEP-induced lung inflammation. However, lung epithelial cells are also a target for DEP and are important producers of KC during lung inflammation. Even maybe (difficult to see) DEP accumulation in lung epithelial cells is observed in Fig-2B. Considering the experimental data presented in the in vivo model, the role of lung epithelium cannot be excluded. This possibility should be at least discussed.
- Analyze of ER stress, UPR pathway and cytokine production in MH-S cells is performed only at RNA level. Authors should perform Western Blots (like in Fig-2) and quantitative dosages in cell supernatants to have information about the protein levels at least for representative parameters.
-Consistency of in vivo and in vitro protocols. Animals are exposed three times during 9 days, however MH-S cells are exposed only one time for 3h. Is there any reason for this? Doses in vivo are equivalent to those in vitro?
-Intracellularly ROS production. % of increase are modest. Authors should include a positive control (H2O2 ?) to illustrate the magnitude of the respond. Moreover, authors should use a ROS inhibitor (e.g NAC) to block DEP-induced production. It will be interesting to study if in the presence of a ROS inhibitor ER stress and inflammation induced by DEP are attenuated.
Minor points
-In printed version of Fig-2 it is difficult to appreciate arrows in panel B. I will suggest to show a representative higher power magnification field. Indicate in the figure legend that mice were treated 3 times and killed at day 9.
-Results section. At the beginning of point 2.4 authors should indicate why they focus in AM based on in vivo data presented in point 2.3
- Authors use a MH-S cell line as a model of AM. This should be indicated at least once in the results section and discussed during discussion section.
-Materials and methods section. For DEP instillation in mice reference 21 is cited. Is this correct?
Author Response
Manuscript ID: molecules-992748
Responses to the Reviewers' Comments
We appreciate very much the editor and the reviewers for the constructive comments. We also thank the editor and the reviewers for the effort and time put into the review of the manuscript. Each comment has been carefully considered point by point and responded. Responses to the reviewers and changes in the revised manuscript are as follows. Thank you for your consideration. I am looking forward to your positive response.
# Reviewer 2: Thank you for your thoughtful and thorough review of our manuscript.
Comments and Suggestions for Authors:
In this manuscript Kim et al. analyze the ER stress and inflammatory responses in a mouse model of diesel exhaust particulate (DEP) exposure (3 times within 9 days). Moreover, they study the induction of inflammation, ER stress and ROS in a murine alveolar macrophage cell line (MH-S). They show that DEP induces neutrophil recruitment and upregulation of ER stress markers in the lungs of DEP-treated mice. Stimulation of MHS cells with DEP induces the expression of ER stress markers, proinflammatory cytokines (including CXCL-1/KC) and production of ROS. The study is interesting because is focused in the understanding of the mechanisms explaining the adverse effects of air pollution on the respiratory system. However, some results are overestimated and final conclusions need to be supported with additional experimental evidences.
Major comments:
1. Title tends to overestimate data present in the paper. Data do not demonstrate that AM present in the lungs of DEP-treated mice are responsible of the neutrophilic inflammation observed. They do not show neither a link between levels of KC in BALF (not showed) and lung inflammation. Data do not show that KC expression is mediated by ER stress in MH-S cells. In line with the previous point, authors should measure levels of KC in the BALF of DEP treated mice in agreement with in vitro data and in order to reinforce the main hypothesis. To improve the hypothesis of the study, could be possible to isolate AM from BALF of treated mice and analyze KC and ER stress markers expression?
Response: Thank you for your comment. We tried to improve the hypothesis of the current study as reviewer’s comments. We added the result of inflammatory cytokine levels including TNF-α, IL-1β, and IL-17 in BALF obtained from in vivo study (page 3, Figure 2C-3E) as well as the results of ER stress markers (BiP and CHOP), inflammatory cytokines (TNF-α, IL-1β, and TLR4), and CXCL1/KC proteins in in vitro study (page 5, Figure 5; page 6, Figure 7). We do not have direct evidence to link with chemokine CXCL1/KC released in DEP-stimulated AM and the ER stress-mediated neutrophilic lung inflammation in our model, but our results showed that the mRNA and protein level of chemokine CXCL1/KC was upregulated in response to DEP stimulation in cell cultured AM. In fact, our group confirmed CXCL1/KC expression in BALF of DEP-instilled C57BL/6 mice. Despite of different strain, we confirmed that DEP induces neutrophil dominant inflammatory responses via measuring inflammatory cells and protein levels in BALF and analyzing H&E stain sections in lung tissues of DEP-treated C57BL/6 mice. Thus, our results suggest that chemokine CXCL1/KC released in DEP-stimulated AM might, at least in part, influence neutrophilic lung inflammation (page 9, line 297-page 10, line 306).
- Authors focus on the role of AM in DEP-induced lung inflammation. However, lung epithelial cells are also a target for DEP and are important producers of KC during lung inflammation. Even maybe (difficult to see) DEP accumulation in lung epithelial cells is observed in Fig-2B. Considering the experimental data presented in the in vivo model, the role of lung epithelium cannot be excluded. This possibility should be at least discussed.
Response: Thank you for your comment. We identified that DEP-laden AM was significantly observed and expression of ER stress-related proteins including BiP and CHOP was statistically increased in lung tissues obtained from in vivo study. So, we assumed that AM may play an important role in the DEP-induced ER stress and inflammatory responses in current study. However, in fact, primary biological targets of inhaled DEP are cells of the pulmonary epithelium and resident macrophages (1). Airway epithelium is the first line of defense of the respiratory system against environmental stimuli. Respiratory exposure to PM including DEP causes airway epithelial cells damages according to deposit on bronchial epithelium and induces toxicity effects on airway and lung disease through releasing various cytokines including IL-8 and granulocyte-macrophage colony-stimulating factor (GM-CSF), IL-1β, IL-6, IL-11, TNF-α, regulated on activated, normal T cell expressed and secreted (RANTES), intercellular cell adhesion molecule 1 (ICAM-1), vascular cell adhesion molecule 1 (VCAM-1), and E-selectin (2, 3). Our results showed that inflammatory cytokine levels including TNF-α, IL-1β, and IL-17 were increased by DEP instillation in BALF obtained from in vivo study. Therefore, lung epithelial cells might be damaged by DEP instillation. However, in our in vivo study, deposed DEP weren’t observed on epithelial cells and the increased infiltration of DEP-laden AM cells were dominantly observed. In this study, we investigated the mechanism underlying the neutrophilic lung inflammatory response, focusing on ER stress, in early stages of DEP-induced mice using cultured AM to understand DEP-induced negative health effects of in vivo study (page 8, line 240- page 9, line 251).
Reference
- Cohen AJ, Pope III CA. Lung cancer and air pollution Environ Health Perspect, 103 (Suppl 8), 1995, 219-224.
- Dawn M. Cooper1 and Matthew Loxham. Particulate matter and the airway epithelium: the special case of the underground? Eur Respir Rev 2019; 28: 190066. ttps://doi.org/10.1183/16000617.0066-2019
- PETER R. MILLS, ROBERT J. DAVIES, and JAGDISH L. DEVALIA. Airway Epithelial Cells, Cytokines, and Pollutants. AM J RESPIR CRIT CARE MED 1999;160:S38–S43.
- Analyze of ER stress, UPR pathway and cytokine production in MH-S cells is performed only at RNA level. Authors should perform Western Blots (like in Fig-2) and quantitative dosages in cell supernatants to have information about the protein levels at least for representative parameters.
Response: We performed western blots analysis of BiP, CHOP, TNF-α, IL-1β, TLR4, and CXCL1/KC proteins in DEP-stimulated MH-S cells and added our results in Figure 5 (page 5, Figure 5; page 6, Figure 7).
- Consistency of in vivo and in vitro protocols. Animals are exposed three times during 9 days, however MH-S cells are exposed only one time for 3h. Is there any reason for this? Doses in vivo are equivalent to those in vitro?
Response: Actually, equivalent doses of in vivo and in vitro need to be considered. However, it difficult to be equivalent to doses of both because various factors need. We focused on the phenomena observed in in vivo study in the current paper, and tried to set an in vitro model to explain this phenomenon. The inflammatory process can be initiated through pathogens and chemicals. In the early stages of inflammation, neutrophils are an important cell population, dominate the acute inflammation, and set the stage for repair of tissue damage by macrophages. In our in vivo study, we showed that DEP induces neutrophilic lung inflammation during 9 days via analyzing histology, differential cell counts, and inflammatory cytokine levels. Especially, our results showed that DEP-laden AM in H&E section of lung tissues was observed and we focused AM. Therefore, acute inflammation is a short-term process and we selected exposure time (3 h) to investigate the response of AM by DEP in early stages of inflammation.
- Intracellularly ROS production. % of increase are modest. Authors should include a positive control (H2O2 ?) to illustrate the magnitude of the respond. Moreover, authors should use a ROS inhibitor (e.g NAC) to block DEP-induced production. It will be interesting to study if in the presence of a ROS inhibitor ER stress and inflammation induced by DEP are attenuated.
Response: Thank you for your comment. As reviewer’s comments, we checked DCF-DA intensities in live cells using FACS analyzer after H2O2 and NAC treatment in DEP-stimulated AM to be more accurate. Our results showed that antioxidant NAC pretreatment significantly decreased DEP-increased ROS production in AM. We added these results in Figure 8C (page 7). Also, we are interested in the effect of ROS inhibitor on ER stress and inflammatory responses in DEP-stimulated AM and we are planning to study that. If we got interesting results, we will submit another research paper in “Molecules” journal.
Minor points:
- In printed version of Fig-2 it is difficult to appreciate arrows in panel B. I will suggest to show a representative higher power magnification field. Indicate in the figure legend that mice were treated 3 times and killed at day 9.
Response: We revised magnification and arrows in Figure 2B and added “Mice were treated 3 times by DEP and sacrificed at day 9.” in figure legend of Figure 2 as reviewer’s comments (page 3, Figure2, line 98-99).
- Results section. At the beginning of point 2.4 authors should indicate why they focus in AM based on in vivo data presented in point 2.3
Response: We added reasons of focus on AM at the beginning of point 2.3 and 2.4 (page 4, line 110-114; page 4, line 121-124).
- Authors use a MH-S cell line as a model of AM. This should be indicated at least once in the results section and discussed during discussion section.
Response: We mentioned the use of a MH-S cell line as a model of AM in results (page 4, line 121-124) and discussion section (page 8, line 234-236).
- Materials and methods section. For DEP instillation in mice reference 21 is cited. Is this correct?
Response: We checked and revised reference number (page 10, line 324).
Round 2
Reviewer 2 Report
I do not have more comments for the authors